# The Importance of Extracellular Vesicle Screening in Gastric Cancer: A 2024 Update

**DOI:** 10.3390/cancers16142574

**Published:** 2024-07-18

**Authors:** Vasile Bintintan, Claudia Burz, Irena Pintea, Adriana Muntean, Diana Deleanu, Iulia Lupan, Gabriel Samasca

**Affiliations:** 1Department of Surgery 1, Iuliu Hatieganu University of Medicine and Pharmacy, 400006 Cluj-Napoca, Romania; vasile.bintintan@umfcluj.ro; 2Institute of Oncology “Prof. Ion Chiricuta”, 400015 Cluj-Napoca, Romania; cristina.burz@umfcluj.ro; 3Department of Immunology, Iuliu Hatieganu University of Medicine and Pharmacy, 400006 Cluj-Napoca, Romania; nedelea@umfcluj.ro (I.P.); adriana.muntean@umfcluj.ro (A.M.); deleanu@umfcluj.ro (D.D.); 4Department of Molecular Biology, Babes-Bolyai University, 400084 Cluj-Napoca, Romania; iulia.lupan@ubbcluj.ro

**Keywords:** extracellular vesicles, update, gastric cancer

## Abstract

**Simple Summary:**

The incidence of gastric cancer (GC) is increasing. Many people receive an advanced diagnosis of GC. Furthermore, a large number of patients are surgically inoperable and are unresponsive to preoperative chemotherapy. Patients with GC do not currently have an immunotherapy option. Our study’s objective was to provide insight into extracellular vesicles (EVs) and GC. EVs in GC can be studied, as this article explains. EVs screening, which is available for all patients, is an option for combatting GC.

**Abstract:**

Extracellular vesicles, or EVs, are membrane-bound nanocompartments produced by tumor cells. EVs carry proteins and nucleic acids from host cells to target cells, where they can transfer lipids, proteomes, and genetic material to change the function of target cells. EVs serve as reservoirs for mobile cellular signals. The collection of EVs using less invasive processes has piqued the interest of many researchers. Exosomes carry substances that can suppress the immune system. If the results of exosome screening are negative, immunotherapy will be beneficial for GC patients. In this study, we provide an update on EVs and GC based on ongoing review papers and clinical trials.

## 1. Introduction

As the third most common cause of cancer-related fatalities and the fifth most common cancer overall, gastric cancer (GC) is a serious global health concern [1]. Because GC rarely exhibits early symptoms, the majority of cases are discovered after the cancer has progressed. Consequently, there is much interest in enhancing early detection by applying focused preventative techniques [2].

These biomarkers include extracellular vesicles (EVs) and circulating proteins from various fluids (such as plasma, serum, urine, and saliva) [3]. Tumor-generated EVs have the potential to stimulate the growth of tumor cells and modify the tumor microenvironment (TME) and immune response. Conversely, EVs derived from immune cells include substances that control the immune response and can eradicate tumors [4]. Tumor growth and development are characterized by immune escape, which may explain why immunotherapy has not been successful. Tumor cells use a variety of strategies to evade immune cells. For instance, they express PD-L1, which binds to the PD-1 receptor of immune cells to prevent T cells from functioning. Cancer cell-derived EVs can spread PD-L1 and other chemicals through bodily fluids, which can depress the immune system [5]. EVs originate from various sources, each containing unique substances that can lead to multiple biological effects. In nearby and distant microenvironments, tumor-derived EVs facilitate bidirectional communication between tumor and stromal cells. EVs that transport messenger ribonucleic acid (mRNAs), microRNAs (miRNAs), short interfering RNAs, DNA fragments, proteins, and metabolites can influence angiogenesis, alter stroma cell morphology, cause immunosuppression, reorganize the extracellular matrix, and change the metabolic environment of organs. Evidence shows that EVs teach stromal cells in secondary locations how to create microenvironments that facilitate metastasis so that tumor cells can seed themselves [6]. It has been demonstrated that exosomal long noncoding RNAs control gene expression through interactions with chromatin-modifying enzymes or transcription factors, which control gene expression by binding to target mRNAs. Preclinical studies have demonstrated that inhibiting particular exosome chemicals can limit tumor development and spread [7]. Exosomes participate in the metabolism of lipids and control several processes related to neoplasms of the digestive system, such as the exchange of information between cancer cells, the development of a neoplastic microenvironment, and the biological behaviors of neoplasms, such as invasion, metastasis, and resistance to chemotherapy [8]. The goal of our study was to provide an update regarding EVs and GC. To identify the most current significant publications in the PubMed database, we utilized the keywords “extracellular vesicles, gastric cancer”. We focused mainly on new research findings from the period 2023–2024. Because they had nothing to do with our topic, case report articles were excluded from the research.

The primary obstacles to the clinical application of EVs are their heterogeneity and varied physicochemical properties. In this study, we classified EVs based on their size and association with GC.

## 2. Bacterial Extracellular Vesicles (bEVs)

Bacterial extracellular vesicles have dimensions that vary from 20 to 400 nm. Various biomolecules, such as proteins, lipids, nucleic acids, and other components derived from the parent bacterium, are present in bEVs [9]. Through the release of bacterial EVs, bacteria can affect the growth of GI tumors. Tumor progression is directly impacted by the interplay of the intestinal flora, bacterial EVs, and TME, which is crucial for tumor survival. Antitumor medication efficacy can be increased by altering the TME through the action of the intestinal microbiota and bacterial EVs [10].

*H. pylori* produces internal outer membrane vesicles (OMVs) and induces host cells to secrete EVs to accomplish these actions. EVs are produced from host cells infected with *H. pylori* after inflammatory signaling pathways that impact immune cell modification, endothelial dysfunction, apoptosis, cytokine release, the disruption of cellular junctional structures, and cytoskeletal reorganization. However, despite this expanding knowledge, studies on EVs produced from *H. pylori* are still in their infancy in a variety of fields [11]. Gene Ontology analysis was performed on 120 significantly differentially expressed proteins found in the EVs of *H. pylori*-infected cells. Using western blotting, CD63 antigen, heat shock protein (HSP)70, and TSG101 were confirmed to be among these proteins. Moreover, EVs from the human gastric epithelial cell line GES-1 infected with *H. pylori* showed a considerable increase in HSP60 expression. HSP60 expression was abnormally elevated in the gastric mucosa, human gastric adenocarcinoma (AGS) cells, GES-1 cells, and GCs as a result of *H. pylori* infection. Furthermore, HSP60 knockdown induced the overexpression of the apoptosis regulator Bax and inhibited the death of infected cells, as did the expression of the Bcl2 apoptosis regulator [12]. By upregulating the expression of the transforming growth factor (TGF)-ß and interleukin (IL)-10 genes in AGS cells and downregulating the mRNA expression of IL-1β, IL-6, IL-8, and tumor necrosis factor (TNF)-α, EVs (50 μg/mL) and cell-free supernatant (CFS) regulated *H. pylori*-induced inflammation. *Lactobacillus crispatus*-derived EVs and CFS therapy significantly reduced *H. pylori*-induced IL-8 production. It may be suggested that EVs produced by the *L. crispatus* strain RIGLD-1 and its CFS could be used as possible therapeutic agents to combat inflammation caused by *H. pylori* [13].

According to several studies, EVs secreted by *F. nucleatum* are linked to the progression of cancer. *F. nucleatum* contributed to the development of GC through *F. nucleatum*-derived EVs (Fn-EVs), which improved our understanding of how to manipulate *F. nucleatum* to treat GC patients with malignant phenotypes. According to in vivo investigations, Fn-EV treatment was shown to confer oxaliplatin resistance to GC tumors in addition to promoting GC tumor development and liver metastasis [14]. The 14-3-3ε protein has a role in a number of physiological processes, such as innate immunity, intracellular electrolyte balance, signal transduction, cell proliferation, apoptosis, autophagy, cell cycle regulation, and the repolarization of cardiac activity. It also has a major impact on the onset and course of a number of illnesses, including cancer, inflammatory diseases, cardiovascular diseases, and neurodegenerative disorders. In individuals with GC, colonization with *F. nucleatum* increases the risk of portal vein thrombosis. The production of neutrophil extracellular traps is stimulated by *F. nucleatum* infection, leading to the release of EVs carrying 14-3-3ε. By activating phosphoinositide 3-kinase and protein kinase B signaling, these EVs may transport 14-3-3ε to hematopoietic progenitor cells and encourage their development into megakaryocytes [15]. We summarize and describe the bacterial EVs in Table 1.

## 3. Small Extracellular Vesicles (sEVs)

sEVs (exosomes, sEVs) are vesicles with a diameter of 30–150 nm. To reprogram target cells, sEVs release bioactive cargos such as proteins and, more specifically, many nucleic acid classes. These cargos aid in tumor growth, cell motility, angiogenesis, immune evasion, and the creation of premetastatic niches [16].

A major obstacle to the prognosis of patients with GC is lymph node metastasis (LNM). Compared to *Helicobacter pylori*-negative GC patients, *H. pylori*-positive patients have a greater incidence of LNM. In *H. pylori*-positive GC patients, miR-1246 is overexpressed in plasma sEVs, which suggests a poor prognosis. In lymphatic endothelial cells, miR-1246 increases the expression of β-catenin and downstream matrix metalloproteinase-7 while suppressing glycogen synthase kinase-3 beta (GSK3β). MiR-1246 also causes CD8+ T cells to undergo apoptosis and suppresses GSK3β, stabilizing PD-L1. In plasma sEVs, miR-1246 may serve as a new biomarker and therapeutic target for GC-LNM [17].

A material’s ability to tolerate variations in length under lengthwise tension or compression is determined by its Young’s modulus. There was a biomimetic peritoneum built. The biomimetic model allows the peritoneal metastasis (PM) process to recur in vitro and is identical to the genuine peritoneum in terms of composition, internal microstructure, and primary function. This model was used to determine the correlation between the invasiveness of GCs and the mechanical characteristics of small extracellular vesicles (sEVs). The Young’s modulus of sEVs can be used to distinguish between nonmalignant clinical samples (peritoneal lavage) and malignant clinical samples (ascites) by performing nanomechanical studies on sEVs. It was confirmed that sEVs taken from the ascites of patients stimulated mesothelial-to-mesenchymal transition, which in turn promoted peritoneal metastasis. The noninvasive detection of GC peritoneal metastasis and the degree of malignancy may be achieved through the nanomechanical analysis of living sEVs [18].

The prognosis of patients with GC is significantly impacted by neural invasion (NI), which is thought to be a mutually beneficial relationship between nerves and malignancies. Intercellular communication is significantly aided by sEVs. The preferential absorption mechanism of GC-derived sEVs by neuronal cells was identified, and a hitherto unknown function of GC-derived sEV-circRNA in GC-NI has been demonstrated. These findings offer fresh perspectives on sEV-circRNA-based diagnostic and treatment approaches for NI-positive GC patients [19]. We summarize and describe the small EVs in Table 2.

## 4. Membrane Vesicles or MVs

EVs released by bone marrow-derived mesenchymal stem cells (BMSCs) are intimately linked to the growth of tumors. Following *H. pylori* infection, the proliferation, migration, and invasion of BMSC-derived microvesicles (membrane vesicles or MVs, 50–100 nm, the main type of EVs) significantly increased in vitro (all *p* < 0.05). Additionally, they promoted tumor development and metastasis in three different in vivo models: tail vein injection metastasis, intraperitoneal (IP) metastasis in nude mice, and subcutaneous xenograft tumors (all *p* < 0.05). Following *H. pylori* therapy, BMSC-derived MVs showed a substantial upregulation of thrombospondin-2 (THBS2) protein expression (*p* < 0.05). In *H. pylori* infection, THBS2 gene depletion decreases the capacity of BMSC-MVs to promote tumors both in vitro and in vivo. By releasing the THBS2 protein, MVs produced from BMSCs can enhance the growth and metastasis of *H. pylori*-associated GC [20].

## 5. EVs and Other Cells in GC

It has been suggested that mucins secreted from epithelial tumors may contribute to cancer-related thrombosis. The amount of the transmembrane mucin called mucin 1 (MUC1) in plasma from healthy controls (HCs) and patients with colorectal, pancreatic, and gastric cancer with or without venous thromboembolism was measured. MUC1 was detected in EV-free plasma from cancer patients and healthy control samples. The MUC1 levels did not significantly differ between cancer patients with and without venous thromboembolism (VTE). Based on our findings, there was no indication that MUC1 caused VTE in cancer patients. There is no evidence that MUC1 causes venous thromboembolism in cancer patients [21].

UPR induction increased the release of procoagulant into the supernatants of GC cells, while antagonists of UPR receptors, such as protein kinase RNA-like endoplasmic reticulum (ER) kinase and inositol-requiring enzyme 1α, inhibited this release. Tissue factor (TF) synthesis was not increased by UPR induction; instead, TF was induced to localize to the cell surface. Verifying the function of vesicular trafficking, ADP-ribosylation factor 1 knockdown or GBF1 antagonism reduced UPR-induced TF delivery to EVs-TF. UPR activation produces a molecular link between ER stress and cancer-associated thrombosis by increasing vesicular trafficking, which in turn releases prothrombotic EVs-TF [22].

## 6. Exosomes

Exosomes are vesicles with a diameter of 30 to 100 nm. Exosomes float in sucrose gradients at a density of 1.13–1.19 g mL^−1^, with a lipid bilayer separating them [23]. Exosomes, the smallest class of extracellular signaling vesicles released by most cells, can communicate across large distances between different tissues and cell types (Figure 1).

### 6.1. The Biology and Roles

Exosomes are essential for cell-to-cell contact and, via the cargo molecules they carry, aid in the growth of malignancies. Because they can be used as treatment solutions for a wide range of diseases and conditions, their application in regenerative medicine has recently attracted considerable interest. Exosomes are used in many different settings, such as GC treatment. These EVs are a viable therapeutic approach for GC because of their inherent targeting capabilities, stability, and biocompatibility [24]. EVs enhance invasion and metastasis in GC through immune-suppressive pathways, altering the TME, boosting angiogenesis, and mediating epithelial–mesenchymal transition (EMT) and mesothelial–mesenchymal transition. The spread of tumors is a multiphase, intricate process. Comprehensive research into the mechanisms of EVs in GC metastasis is essential since EVs frequently interact through various pathways to promote GC metastasis [25]. The development of novel approaches to targeted therapy would be highly advantageous, as conventional treatments such as radiation, chemotherapy, and surgery are confronted with several disadvantages, such as acquired drug resistance and numerous side effects, which can lead to cancer recurrence and increased morbidity. Exosomes produced from mesenchymal stem cells (MSC-EXs) may be a double-edged sword in certain malignancies. The use of MSC-EXs from the umbilical cord and foreskin as promising sources for increasing the effectiveness of GC treatment was suggested as one possible change and combination option [26]. Exosomes prevent GC cells from adhering to human peritoneal mesothelial cells (HPMCs) and prevent HPMCs from proliferating and migrating. When miR-29b-rich exosomes were subjected to IP every three days, the number of peritoneal metastases (PMs) of the murine GC cell line YTN16P on the mesentery of C57BL/6 mice was significantly decreased. The IP delivery of exosomes expressing miR-29b inhibits the growth of PM in GC [27]. In a patient-derived MET proto-oncogene, receptor tyrosine kinase (MET)-amplified GC model, exosomal MET depletion significantly decreased invasiveness; when paired with MET and/or vascular endothelial growth factor (VEGF)-R2 inhibition, an additive therapeutic effect was induced. In a GC model without MET amplification, the allogeneic transfer of exosomes carrying MET resulted in invasion and angiogenesis. The exosome concentration was greater in MET-amplified patient tissues than in neighboring normal tissues. A potentially effective supplementary tactic against GC could involve modifying the content and production of exosomes [28].

### 6.2. Circular RNA

A particular form of noncoding RNA called circular RNA (circRNA) is extensively enriched in exosomes and plays a role in several pathological processes that are exosome-mediated. Exosomal circRNAs have become important participants in a variety of tumor types. By controlling the EMT, angiogenesis, proliferation, invasion, migration, and metastasis of GC, exosomal circRNAs are essential for the formation of GC [29]. Although circRNAs are involved in the development of GC, it is still unclear exactly how they regulate macrophage function. In addition to acting as carriers of circRNAs, exosomes are essential mediators in the communication process between cancer cells and the TME. Tumor development and M2 macrophage (alternatively activated) polarization are induced by exosome-derived circATP8A1 from GC cell lines (AGS, SGC-7901, HGC-27, and MKN-45) via the circATP8A1/miR-1-3p/STAT6 axis. In vitro and in vivo GC invasion and proliferation were dramatically reduced by circATP8A1 knockdown. circATP8A1 is hence a possible therapeutic target and prognostic biomarker for GC [30]. M2-like macrophages and high mobility group protein B1 (HMGB1) are tightly linked to fundamental aspects of human malignancies, including invasion, metastasis, and the cancer microenvironment. Exosomes with high HMGB1 levels triggered M2-like macrophage polarization and accelerated GC development, contrary to encouraging the proliferation of GC cells. Thus, the novel approach by which HMGB1 drives the progression of GC may offer fresh perspectives on enhancing the effectiveness of cancer immunotherapy [31]. Exosomes are important mediators of communication between tumor cells and tumor-associated macrophages because they contain a wide range of physiologically active substances, including lipids, proteins, mRNAs, and noncoding RNAs. These chemicals can be exchanged by exosomes, which can significantly alter the TME and, in turn, affect how quickly the tumor progresses [32].

Circ50547 is highly expressed in serum exosomes and GC tissues. Exosomal circ50547 has greater diagnostic value than serum circ50547. The overexpression of circ50547 improved GC cell proliferation, migration, invasion, stemness, and drug resistance; moreover, circ50547 knockdown had the opposite effect. To control the expression of hepatocyte nuclear factor 1 homeobox B and accelerate the development of GC, Circ50547 serves as a sponge for miR-217 [33]. EVs and circRNAs in tumors are essential for the malignant phenotype of tumor cells. It is unknown how EV-delivered hsa_circ_0090081 affects GC and what its clinical implications are. The expression levels of eukaryotic translation initiation factor 4A3 (EIF4A3) and hsa_circ_0090081 were elevated in GC, and hsa_circ_0090081 was linked to a poor prognosis. The data showed that the inhibition of hsa_circ_0090081 limited the migration, invasion, and proliferation of GC cells. Consequently, EIF4A3 promoted GC development via hsa_circ_0090081 [34]. A total of 13,229 upregulated and 7079 downregulated mRNAs were found in serum EVs by RNA sequencing. According to Gene Ontology and Kyoto Encyclopedia of Genes and Genomes analyses, certain mRNAs are linked to the development and metastasis of tumors. Ten of these genes were chosen based on the following standards: *p* < 0.05. In patients with GC, nuclear receptor-binding SET domain protein 1 (NSD1) was increased, and F-box-only protein 7 (FBXO7) was downregulated in comparison to healthy controls. Together, these two mRNAs had an area under the receiver operating characteristic (ROC) curve of 0.84, a sensitivity of 78%, and a specificity of 92%. The malignant tumor stage, distal metastasis status, and tumor size were also linked to NSD1 and FBXO7 expression. The prognosis was strongly correlated with NSD1 expression. According to follow-up data, FBXO7 was strongly correlated with prognosis [35]. The suppression of hsa_circ_0014606 resulted in increased apoptosis, decreased migration and invasion, a decreased tumor size in mice, and an inhibited proliferation of GC cells. Heterogeneous nuclear ribonucleoprotein C (HNRNPC) expression and miR-514b-3p expression were positively correlated and negatively correlated, respectively, with the expression of hsa_circ_0014606. Through miR-514b-3p targeting the gene HNRNPC, hsa_circ_0014606 indirectly promoted cancer, and this work offers a novel therapeutic target for GC [36]. The tumor size, miR-21-5p, and miR-27a-3p showed a strong correlation. Moreover, following surgery, individuals with GC had significantly lower serum levels of miR-21-5p and miR-26a-5p. Consequently, it was discovered that serum miR-21-5p and miR-26a-5p have the potential to be useful indicators for both the diagnosis and prognosis of GC [37]. It was discovered that miR-410-3p is markedly downregulated in GC. The proliferation, migration, and invasion of GC cells were reduced by the overexpression of miR-410-3p. MiR-410-3p mimics increased cell adhesion. In primary GC, miR-410-3p targeted HMGB1. Exosomal miR-410-3p was expressed at a significantly greater levels in cell culture media than in the body. Exosomes derived from AGS or BCG23 (cell culture media from the stomach) altered the endogenous expression of miR-410-3p in MKN45 cells. In cases of primary GC, miR-410-3p acted as a tumor suppressor [38].

### 6.3. Therapeutic Applications

Melatonin controlled the exosomal miR-27b-3p-ADAMTS5 pathway, which inhibited the growth of GC. Melatonin is a potentially effective treatment medication for people diagnosed with GC [39]. For GC with PM, IP chemotherapy combined with paclitaxel (PTX) is thought to be a promising therapeutic option. IP exo-miR-493 downregulates Mitotic Arrest Deficient 2 Like 1 in GCs with PM, contributing to chemoresistance to PTX. Exo-miR-493 could be a promising target for therapy as well as a biomarker for the prognosis and chemoresistance of GC patients with PM [40]. We summarize and describe the circRNA exosomes in Table 3.

## 7. Exosomes Derived from Cancer-Associated Fibroblasts (CAFs)

Cancer-associated fibroblasts, or CAFs, are a significant feature of the TME. They have been identified as agents involved in tumor immune evasion and are involved in several aspects of cancer progression. The function of CAFs in different kinds of tumors, including GC, has recently garnered increasing attention. The top five genes in the field of CAFs that have attracted the most attention in terms of study are TGF-ß1, IL-6, TNF, tumor protein P53, and VEGF-A. The top 20 most investigated genes were primarily linked to the hypoxia-inducible factor-1 and Toll-like receptor signaling pathways according to Kyoto Encyclopedia of Genes and Genomes enrichment analysis [41]. How exosomal miR-29b-1-5p generated from CAFs affects GC cells was investigated. GC tissues displayed a vascular mimicry (VM) shape, the downregulation of immunoglobulin domain-containing 1 (VSIG1) and zonula occluden-1 (ZO-1), and the overexpression of miR-29b-1-5p. When targeted by miR-29b-1-5p, VSIG1 interacts with ZO-1. Exosomal miR-29b-1-5p inhibitors generated from CAFs inhibited VM formation, migration, invasion, and survival in GC cells while promoting death. In vitro and in vivo, the miR-29b-1-5p inhibitor reduced the levels of VE-cadherin, N-cadherin, and vimentin while increasing those of VSIG1, ZO-1, and E-cadherin. However, shVSIG1 partially reversed these effects. By upregulating VSIG1/ZO-1 expression, the downregulation of CAF-derived exosomal miR-29b-1-5p hindered GC carcinogenesis and the VM structure in vivo. Therefore, GC development is inhibited by downregulating the exosomal miR-29b-1-5p generated from CAFs via the VSIG1/ZO-1 axis [42]. Compared with those of healthy individuals and patients with benign gastric disorders, the serum exosomal long intergenic nonprotein coding RNA 691 (LINC00691) level of GC patients was significantly greater, and this difference was linked to the clinicopathology of GC patients. The induced fibroblasts acquired the characteristics of CAFs when the normal fibroblasts (NFs) were treated with GC exosomes because of the marked increase in LINC00691, the noticeable enhancement of cell proliferation and migration, and the promotion of GC cell proliferation and invasion. Ruxolitinib, an inhibitor of the Janus kinase 2 (JAK2)/signal transducer and activator of the transcription 3 (STAT3) signaling pathway, and LINC00691 knockdown substantially eliminated the effects on NFs in exosome-containing GC cell supernatants. As a possible diagnostic biomarker for GC, exosomal LINC00691 stimulated NFs to acquire the characteristics of CAFs reliant on the JAK2/STAT3 signaling pathway [43].

## 8. Options for the Implementation of EV Screening in GC

Significant technological advancements in recent years have made it possible to profile these short noncoding RNAs in EVs in detail because of their potential as cancer biomarkers. These indicators, in particular, are noninvasive liquid biopsy markers that can be used to detect a variety of malignancies, including gastrointestinal (GI) tumors [44]. Despite recent developments in molecular understanding, many of these patients’ prognoses have not improved much with the availability of therapeutic options. The use of liquid biopsy in GC may prove to be quite beneficial as a noninvasive technique for prompt diagnosis and in improving patient care and results [45]. To identify microRNAs in GC-derived EVs, a unique electrochemiluminescence (ECL) biosensor was created. EVs from the ascites of clinical GC patients were removed and examined. The findings demonstrated the diagnostic utility of the built-in biosensor for detecting GC peritoneal metastases [46].

It is yet unknown how GINS complex subunit 1 (GINS1) and DS-cell cycle-dependent protein 1 (DSCC1) function in GC. Protective factors include the core genes (TRIP13, CHEK1, DSCC1, and GINS1), whose expression tends to decrease as risk scores increase. TRIP13, CHEK1, DSCC1, and GINS1 were linked to GCs, gastric illnesses, tumors, inflammation, and necrosis according to a comparative toxicogenomic database analysis. In GC, DSCC1 and GINS1 are abundantly expressed. The prognosis is worse for patients with higher DSCC1 and GINS1 expression levels [47]. TENM2 (area under the curve, AUC = 0.982), CD36 (AUC = 0.974), and CD36-ITGA1 (AUC = 0.971) are just a few of the eight differentially expressed protein combinations (DEPCs) and five differentially expressed proteins (DEPs) that have shown excellent performance (AUC > 0.900) in pancancer diagnosis. Using DEPs (AUC = 0.981) or DEPCs (AUC = 0.965), this classification model was able to distinguish between cancer patients and controls with accuracy. The accuracy of the classification model using DEPCs (85–92%) was greater than that using DEPs (78–84%) for differentiating one malignancy from the other four. It was discovered that there were heterogeneous EV clusters abundant in cancer patients who correlated with the initiation and progression of the tumor [48].

The most prevalent immune cells in human circulation, neutrophils, are essential for the development of tumors. Neutrophil-derived exosomes, or Neu-Exos, are vital to the development of disease because they are rich in bioactive chemicals. Under ideal circumstances, dual antibody-assisted fluorescent Dynabeads achieved a recovery rate of 81% and a detection limit of 7.8 × 105 Neu-Exo particles/mL. According to the ROC curve, GC patients could be easily distinguished from HCs based on the quantity of CD66b+ Neu-Exos (AUC > 0.8). One of the most highly differentially expressed miRNAs in Neu-Exos was identified to be miR-223-3p, which also has good diagnostic value for GC. When used to distinguish GC patients from HCs and individuals with benign gastric disorders, droplet digital PCR dramatically increased the diagnostic efficacy (AUC > 0.9) [49]. The use of EVs for the transport of miRNAs has recently spurred enormous advancements in cancer therapies. To evaluate the treatment benefits in real time, it is currently challenging to observe the hybridization between miRNA and its target mRNA in vivo. sEVs encapsulating a nanoscale fluorescent “off–on” complex were created for monitoring and assessing the effectiveness of gene therapy in human GC cells and murine xenograft tumor models in real time. π stacking (π–π stacking) occurs between the fluorescent tag conjugated to the tumor suppressor miR-193a-3p, whose signals remain off while binding to graphene quantum dots (GQDs), and the GQDs form the complex. Through increased cellular absorption, GQD/Cy5-miR@sEVs dramatically accelerated cancer cell apoptosis both in vitro and in vivo. Gene therapy for malignancies can be effectively and efficiently facilitated by GQDs/Cy5-miR@sEVs [50]. When sEVs are produced from different cell lines, they exhibit homologous blood circulation and tumor accumulation, but sEVs tagged with different fluorescent dyes exhibit unique tumor accumulation properties. Therefore, single labelling of the sEV membrane should be avoided in future studies when examining the biological fate of sEVs, even though fluorescence imaging is still a reliable method for tracking sEVs [51].

A trimode aptasensor comprising magnetically regulated photothermal, colorimetric, and fluorescence controls was created for exosomes produced from human GC cells (SGC-7901). Reduced absorbance, temperature, and fluorescence intensity were associated with increasing exosome concentrations. Using human serum samples, the trimode biosensor showed a satisfactory ability to distinguish between patients with GC and healthy people [52].

## 9. Conclusions

The two most studied types of EVs in the CG are bEVs and exosomes. Due to the heterogeneity of EVs, research on EVs in GC is possible. The potential applications of exosome screening in GC patients include early diagnosis and follow-up. As these GC patients are often diagnosed in advanced stages, the detection and screening of EVs could improve the diagnosis of GC and represent a step forward in gastroenterology medical clinics. We recommend screening for EVs in GC in the clinic, and funding should be available for screening.

## Figures and Tables

**Figure 1 cancers-16-02574-f001:**
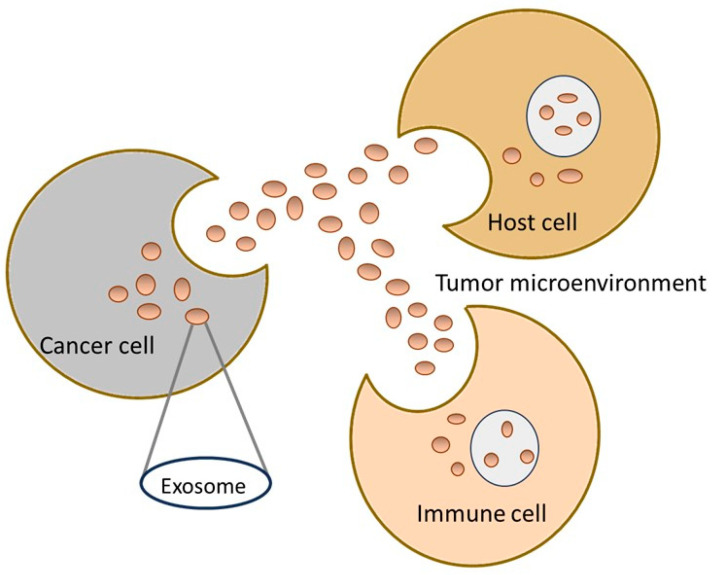
Exosomes communicate across large distances between different cell types.

**Table 1 cancers-16-02574-t001:** A recapitulative table detailing the bacterial EVs.

The Experimental Design	The Kind of Approach	The Main Results	References
bEVs and gut microbiota can alter TME to improve the efficacy of anticancer medications.	The biological targets that the gut microbiota use to control TME	Possible uses of bEVs in the prognosis, diagnosis, and therapy of tumors.	[10]
OMVs are produced by *H. pylori*.	OMVs from *H. pylori* are essential in determining the course of ensuing immunopathological reactions.	Potential use of OMVs as therapeutic targets, biomarkers, and delivery systems for certain medications.	[11]
EVs extracted from *H. pylori*-infected cells underwent a thorough proteome study, and the function of EV-derived HSP60 was investigated.	EVs were assessed by western blotting, transmission electron microscopy, and nanoparticle tracking analysis.	The results highlight the role that EV-derived HSP60 plays in the pathogenesis of illnesses linked to *H. pylori*.	[12]
AGS cells were used to study the immunomodulatory effects of *L. crispatus*-derived EVs and CFS on *H. pylori*-induced inflammatory responses.	Using sodium dodecyl sulfate–polyacrylamide gel electrophoresis, the protein content of EVs generated from *L. crispatus* was also assessed.	It may be suggested to use the EVs produced by the *L. crispatus* strain RIGLD-1 and its CFS as possible therapeutic agents to combat inflammation caused by *H. pylori*.	[13]
From 30 GC patients, the tumor and surrounding tissues were surgically removed.	*F. nucleatum* secretes EVs, which aid in the advancement of cancer.	It was discovered that tumor samples had a high expression level of *F. nucleatum*.	[14]
In patients with GC, *F. nucleatum* colonization is a contributing factor to the development of portal vein thrombosis.	Fluorescence in situ hybridization and quantitative PCR were used to look for *F. nucleatum* in the tumor and surrounding nontumor tissues in 91 GC patients.	An infection with *F. nucleatum* stimulates the development of NETs, which release EVs carrying 14-3-3ε. Through the stimulation of PI3K-Akt signaling, these EVs may transport 14-3-3ε to HPCs and encourage their differentiation into MKs.	[15]

**Table 2 cancers-16-02574-t002:** A recapitulative table detailing small EVs.

The Experimental Design	The Kind of Approach	The Main Results	References
Compared to GC patients who are *H. pylori*-negative, patients with *H. pylori* positivity have a greater incidence of LNM but uncertainty surrounds the fundamental mechanism.	*H. pylori*-positive GC patients had overexpressed miR-1246 in their plasma sEVs, which suggests a poor prognosis.	In plasma sEVs, miR-1246 may provide a new biomarker and therapeutic target for GC-LNM.	[17]
A peritoneum mimicking biology was built.	The biomimetic model allows for the in vitro recurrence of the peritoneal metastatic process.	It is possible to distinguish between malignant clinical samples and nonmalignant clinical samples using the Young’s modulus of sEVs.	[18]
It is unknown if sEVs mediate GC-NI. sEVs release inhibitor lowers GC cells’ NI potential.	The prognosis of patients with GC is significantly impacted by NI, which is thought to be the mutually beneficial relationship between nerves and malignancies.	This study reveals the mechanism by which neuronal cells preferentially absorb GC-derived sEVs and highlights a hitherto unknown function of GC-derived sEV-circRNA in GC-NI.	[19]

**Table 3 cancers-16-02574-t003:** A recapitulative table detailing the circRNA exosomes.

The Experimental Design	The Kind of Approach	The Main Results	References
Transwell assays, Cell Counting Kit—8, and colony formation were used to investigate the role of hsa_circ_0050547.	Examine the role that exosomal circRNAs play in the detection and advancement of GC.	Circ50547, a novel kind of GC-derived exosomal circRNA, was identified.	[33]
To determine the amounts of EIF4A3 and hsa_circ_0090081 in GC tissues, qRT–PCR was used.	The aggressive nature of tumor cells is dependent on EVs and circRNA in malignancies.	EIF4A3 improved hsa_circ_0090081, enabling it to promote GC development.	[34]
Four patients with GC and four healthy controls had their serum exosomes examined for the presence of mRNAs.	EVs containing mRNAs in serum are useful noninvasive biomarkers for a number of cancer types; however, little is known about their potential as GC biomarkers.	In serum EVs, NSD1 and FBXO7 play significant roles in GC and may be helpful indicators for both diagnosis and prognosis.	[35]
By using ultracentrifugation, exosomes were separated from the serum of both healthy people and GC patients. The expression of hsa_circ_0014606 in each exosome was then examined.	It is unknown how hsa_circ_0014606 within exosomes contributes to the development of GC.	hsa_circ_0014606 had an indirect pro-cancer effect by targeting the gene HNRNPC with miR-514b-3p.	[36]
MicroRNAs that were differentially expressed were chosen at random and confirmed using quantitative real-time polymerase chain reaction (RT-QC) and reverse transcription.	Circular EVs could prove to be effective biomarkers for a quick and noninvasive diagnosis.	Serum miR-21-5p and miR-26a-5p have been shown to have potential as diagnostic and prognostic indicators for GC.	[37]
RT–qPCR was used to assess the endogenous expression of miR-410-3p in tissue samples and cell lines as well as the expression of exosomal miR-410-3p in a cell culture medium.	Exosome functions in controlling miR-410-3p expression in GC were investigated.	Exosomes originating from the initial site may control the endogenous expression of miR-410-3p in a distant region.	[38]
A total of 34 exosomal miRNAs in AGS cells were screened using exosome small RNA sequencing (sRNA-seq) to identify significant alterations before and after melatonin administration.	The impact of melatonin-treated GC cells’ exosomal miRNAs on GC was investigated.	Melatonin controlled the exosomal miR-27b-3p-ADAMTS5 pathway, which inhibited the growth of GC.	[39]
When compared to exosomes from patients who responded to IP therapy with PTX, MKN45 cells that were grown with intraperitoneal exosomes from patients who did not respond to IP therapy with PTX showed resistance to PTX (*p* = 0.002).	IP-chemo sensitivity and intraperitoneal exosomes, specifically exosomal micro-RNAs (exo-miRNAs), were examined in relation to each other.	Intraperitoneal exo-miR-493 downregulates MAD2L1 in GC with PM, contributing to chemoresistance to PTX.	[40]

## Data Availability

Not applicable.

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
