# Peer review of "The Importance of Extracellular Vesicle Screening in Gastric Cancer: A 2024 Update"

_cancers, 2024, doi:10.3390/cancers16142574_

Round 1

Reviewer 1 Report

Comments and Suggestions for Authors

The review entitled “The Importance of Extracellular Vesicle Screening in Gastric 2 Cancer: A 2024 Update” concerns a very important topic, that is the emerging role(s) of EVS in GC pathology and the potential use of some EV cargo molecules in clinics at different levels.

There are several parts that may be improved. First of all, authors should focus on the only works performed in GC, thus avoiding citations about other systems (ex. plants) or pathologies.

Helicobacter pylori infections should be described when the work is focused on GC. At line 66, the sentence “There was a biomimetic peritoneum built” should be better explained since the further biomimetic model is argue to be understood. 

In §3, bacterial EVs are cited to affect the growth of gastrointestinal tumours. EVs from milk are cited. Moreover, it is cited that “At the final stage of the intestinal phase, intact EVs are discovered”. It is not clear the link between milk-EVs and it is argue to understand what is the “final stage of the intestinal phase”, and, most importantly, if all these details can provide useful information to the topic of the review (GC screening).

Line 83: are internal membrane vesicles abbreviated with OMVs?

Line 92: proteins and not genes.

Lines102-104 refer to a work entitled “Effect of In Vitro Enzyme Digestion and Bile Treatment on Milk Extracellular Vesicles Stability”. Is Lactobacillus crispatus resident in stomach and has it be proven to produce EVs  in human stomach?

In §4, the cited “transmembrane mucin called mucin 1 (MUC1)” overexpressed in several human cancers where it is related to venous thromboembolism, is it overexpressed in terms of protein in GC too ?

In §5 the role of EVS/exosomes in communication process is redundant since it is abundantly described in the review. Is it known which kind of GC cells produce exosome-derived circATP8A1 ? Is that an in vitro study ? Is Ficolin B (Fcn B) produced in human stomach or only on a mouse model ?

The §6 should be written again focusing on the only data available for GC.

The §Conclusion should not only cite the potential application of exosomes in target therapy but also in early diagnosis and follow-up. What are the real expectations for the use of EV molecules in Clinics and GC screening ?

In some parts, the review is very hard to be read.

A summary illustration may help.

Similarly a recapitulative table detailing the different EV molecules, the experimental design (GC cell culture; in vivo mouse model; biopsies), the kind of approach, the main results may help the reader to better follow the descriptive part.

Authors should focus on the only works on GC.

They should avoid abbreviations where they are only cited once.

Some repetition about the roles played by EVs and exosomes should be avoided.

Comments on the Quality of English Language

The review entitled “The Importance of Extracellular Vesicle Screening in Gastric 2 Cancer: A 2024 Update” concerns a very important topic, that is the emerging role(s) of EVS in GC pathology and the potential use of some EV cargo molecules in clinics at different levels.

There are several parts that may be improved. First of all, authors should focus on the only works performed in GC, thus avoiding citations about other systems (ex. plants) or pathologies.

Helicobacter pylori infections should be described when the work is focused on GC. At line 66, the sentence “There was a biomimetic peritoneum built” should be better explained since the further biomimetic model is argue to be understood. 

In §3, bacterial EVs are cited to affect the growth of gastrointestinal tumours. EVs from milk are cited. Moreover, it is cited that “At the final stage of the intestinal phase, intact EVs are discovered”. It is not clear the link between milk-EVs and it is argue to understand what is the “final stage of the intestinal phase”, and, most importantly, if all these details can provide useful information to the topic of the review (GC screening).

Line 83: are internal membrane vesicles abbreviated with OMVs?

Line 92: proteins and not genes.

Lines102-104 refer to a work entitled “Effect of In Vitro Enzyme Digestion and Bile Treatment on Milk Extracellular Vesicles Stability”. Is Lactobacillus crispatus resident in stomach and has it be proven to produce EVs  in human stomach?

In §4, the cited “transmembrane mucin called mucin 1 (MUC1)” overexpressed in several human cancers where it is related to venous thromboembolism, is it overexpressed in terms of protein in GC too ?

In §5 the role of EVS/exosomes in communication process is redundant since it is abundantly described in the review. Is it known which kind of GC cells produce exosome-derived circATP8A1 ? Is that an in vitro study ? Is Ficolin B (Fcn B) produced in human stomach or only on a mouse model ?

The §6 should be written again focusing on the only data available for GC.

The §Conclusion should not only cite the potential application of exosomes in target therapy but also in early diagnosis and follow-up. What are the real expectations for the use of EV molecules in Clinics and GC screening ?

In some parts, the review is very hard to be read.

A summary illustration may help.

Similarly a recapitulative table detailing the different EV molecules, the experimental design (GC cell culture; in vivo mouse model; biopsies), the kind of approach, the main results may help the reader to better follow the descriptive part.

Authors should focus on the only works on GC.

They should avoid abbreviations where they are only cited once.

Some repetitions about the roles played by EVs and exosomes should be avoided.

Author Response

The review entitled “The Importance of Extracellular Vesicle Screening in Gastric 2 Cancer: A 2024 Update” concerns a very important topic, that is the emerging role(s) of EVS in GC pathology and the potential use of some EV cargo molecules in clinics at different levels.

1.There are several parts that may be improved. First of all, authors should focus on the only works performed in GC, thus avoiding citations about other systems (ex. plants) or pathologies.

We cut studies related to plants

“Plant-derived EVs have been the subject of much research in recent years due to growing interest in their possible applications as delivery systems for medicinal substances. Moreover, research has demonstrated that EVs produced by the gut microbiota of the host can travel via the bloodstream and enter distant organs and tissues (8). EVs from Kaempferia parviflora (KPEVs) have been described as promising nanovesicles for drug delivery. KPEVs can transport clarithromycin (CLA). They were more effective than free CLA at reducing the inflammation of gastric cells caused by Helicobacter pylori infection. The development of KPEVs-CLA as a model for nanodrug delivery for the treatment of H. pylori may be extended to other plant EVs or combined with additional cancer-fighting medications for the treatment of GC (9).”

We focused only on gastric cancer.

2.Helicobacter pylori infections should be described when the work is focused on GC. At line 66, the sentence “There was a biomimetic peritoneum built” should be better explained since the further biomimetic model is argue to be understood. 

We explained it better in text:

‘There was a biomimetic peritoneum built. The biomimetic model allows the peritoneal metastasis process to recur in vitro and is identical to the genuine peritoneum in terms of composition, internal microstructure, and primary function. This model was used to determine the correlation between the invasiveness of GC and the mechanical characteristics of sEVs. Young's modulus of sEVs can be used to distinguish between nonmalignant clinical samples (peritoneal lavage) and malignant clinical samples (ascites) by performing nanomechanical studies on sEVs. It was confirmed that the sEVs taken from the ascites of the patients stimulated the mesothelial-to-mesenchymal transition, which in turn promoted peritoneal metastasis. In conclusion, the noninvasive detection of GC's peritoneal metastases and malignant degree may be achieved through the use of nanomechanical analysis of living sEVs.”

3.In §3, bacterial EVs are cited to affect the growth of gastrointestinal tumours. EVs from milk are cited. Moreover, it is cited that “At the final stage of the intestinal phase, intact EVs are discovered”. It is not clear the link between milk-EVs and it is argue to understand what is the “final stage of the intestinal phase”, and, most importantly, if all these details can provide useful information to the topic of the review (GC screening).

We deleted this paragraph

4.Line 83: are internal membrane vesicles abbreviated with OMVs?

We corrected:

H. pylori produce internal outer membrane vesicles (OMVs)”

5.Line 92: proteins and not genes.

We corrected:

“Using western blotting, CD63 antigen, 70 kilodalton heat shock protein (HSP-70), and tu-mor susceptibility gene 101 (TSG101) were confirmed to be among these proteins.”

6.Lines102-104 refer to a work entitled “Effect of In Vitro Enzyme Digestion and Bile Treatment on Milk Extracellular Vesicles Stability”. Is Lactobacillus crispatus resident in stomach and has it be proven to produce EVs  in human stomach?

We corrected:

Fakharian F, Sadeghi A, Pouresmaeili F, Soleimani N, Yadegar A. Anti-inflammatory effects of extracellular vesicles and cell-free supernatant derived from Lactobacillus crispatus strain RIGLD-1 on Helicobacter pylori-induced inflammatory response in gastric epithelial cells in vitro. Folia Microbiol (Praha). 2024 Feb 3. doi: 10.1007/s12223-024-01138-3.”

- the bibliographic reference was wrong!

7.In §4, the cited “transmembrane mucin called mucin 1 (MUC1)” overexpressed in several human cancers where it is related to venous thromboembolism, is it overexpressed in terms of protein in GC too ?

We explained it better in text:

“It has been suggested that mucins secreted from epithelial tumors may contribute to cancer-related thrombosis. The amount of transmembrane mucin called mucin 1 (MUC1) in plasma from healthy controls and patients with patients with stomach, colorectal or pancreatic cancer with or without venous thromboembolism, has been measured. There was no evidence that MUC1 causes venous thromboembolism in cancer patients [21].”

8.In §5 the role of EVS/exosomes in communication process is redundant since it is abundantly described in the review. Is it known which kind of GC cells produce exosome-derived circATP8A1 ? Is that an in vitro study ?

We explained it better in text!

Is it known which kind of GC cells produce exosome-derived circATP8A1 ?

  • “GC cells (AGS, SGC-7901, HGC-27, and MKN-45)”

Is that an in vitro study ?

  • “In vitro and in vivo GC invasion and proliferation were dramatically reduced by circATP8A1 knockdown.”

9.Is Ficolin B (Fcn B) produced in human stomach or only on a mouse model ?

We deleted this study because induced lung injury:

– Wu X, Jiang Y, Li R, Xia Y, Li F, Zhao M, Li G, Tan X. Ficolin B secreted by alveolar macrophage exosomes exacerbates bleomycin-induced lung injury via ferroptosis through the cGAS-STING signaling pathway. Cell Death Dis. 2023 Aug 30;14(8):577. doi: 470 10.1038/s41419-023-06104-4.

10.The §6 should be written again focusing on the only data available for GC.

We rewrite and focused only on GC.

“Cancer-associated fibroblasts, or CAFs, are a significant feature of the tumor microen-vironment. They have been identified as agents of tumor immune evasion and have been involved in several aspects of the progression of cancer. The function of CAFs in different kinds of tumors, including GC, has garnered more focus lately. The top 5 genes in the field of CAFs that have attracted the most attention in terms of study is transforming growth factor beta 1 (TGFB1), IL-6, TNF, tumor protein P53 (TP53), and vascular endothelial growth factor A (VEGFA). The top 20 most investigated genes were primarily linked to the hypoxia-inducible factor (HIF)-1 and Toll-like receptor signaling pathways according to KEGG enrichment analysis [45]. How exosomal miR-29b-1-5p generated from CAFs affects GC cells was investigated. GC tissues displayed a vascular mimicry (VM) shape, down-regulation of immunoglobulin domain-containing 1 (VSIG1) and zonula occluden-1 (ZO-1), and overexpression of miR-29b-1-5p. When targeted by miR-29b-1-5p, VSIG1 in-teracts with ZO-1. Exosomal miR-29b-1-5p inhibitors generated from CAFs inhibited VM formation, migration, invasion, and survival in GC cells while promoting death. In vitro and in vivo, the miR-29b-1-5p inhibitor reduced the levels of VE-cadherin, N-cadherin, and vimentin while increasing those of VSIG1, ZO-1, and E-cadherin. However, shVSIG1 partially reversed these effects. By upregulating VSIG1/ZO-1 expression, downregulation of CAF-derived exosomal miR-29b-1-5p hindered GC carcinogenesis and VM structure in vivo. Therefore, GC development is inhibited by downregulating the exosomal miR-29b-1-5p generated from CAFs via the VSIG1/ZO-1 axis [46]. Compared with those of healthy individuals and patients with benign gastric disorders, the serum exosomal long intergenic nonprotein coding RNA 691 (LINC00691) level of GC patients was significantly greater, and this difference was linked to the clinicopathology of GC patients. The induced fibroblasts acquired the characteristics of CAFs when the normal fibroblasts (NFs) were treated with GC exosomes because of the marked increase in LINC00691, the noticeable enhancement of cell proliferation and migration, and the promotion of GC cell prolifera-tion and invasion. Ruxolitinib, an inhibitor of the Janus kinase 2 (JAK2)/signal transducer and activator of transcription 3 (STAT3) signaling pathway, and LINC00691 knockdown substantially eliminated the effects on NFs in exosome-containing GC cell supernatants. As a possible diagnostic biomarker for GC, exosomal LINC00691 stimulated NFs to ac-quire the characteristics of CAFs reliant on the JAK2/STAT3 signaling pathway [47].”

11.The §Conclusion should not only cite the potential application of exosomes in target therapy but also in early diagnosis and follow-up. What are the real expectations for the use of EV molecules in Clinics and GC screening ?

We rewrote THE CONCLUSIONS:

  1. Conclusions

The two most studied types of EVs in the CG are bEVs and exosomes. Due to the heterogeneity of EVs, research on EVs in GC is possible. The potential applications of exosome screening in GC patients include early diagnosis and follow-up. As these GC patients are often diagnosed in advanced stages, the detection and screening of EVs could improve the diagnosis of GC and represent a step forward in gastroenterology medical clinics. We recommend screening for EVs in GC in the clinic, and funding should be available for screening.

12.In some parts, the review is very hard to be read.

We corrected all the mistakes!

VERIFICAM ENGLEZA LA SFARSIT

13.A summary illustration may help.

We have introduced figure 1.

14.Similarly a recapitulative table detailing the different EV molecules, the experimental design (GC cell culture; in vivo mouse model; biopsies), the kind of approach, the main results may help the reader to better follow the descriptive part.

We have introduced Table 1, Table 2 and Table 3

15.Authors should focus on the only works on GC.

We focused only on GC

16.They should avoid abbreviations where they are only cited once.

We eliminated abbreviations where they are only cited once.

17.Some repetition about the roles played by EVs and exosomes should be avoided.

We avoided some repetition about the roles played by EVs.

Reviewer 2 Report

Comments and Suggestions for Authors

In this review paper, the authors reviewed EVs of gastric cancers. The review may help to understand the function and mechism of gastric cancer EVs but there are some suggestions for improving the work, it cannot be accepted at this state.

 1. As a review paper, it would be beneficial to include summaries with figures and tables to assist readers in better grasping the key points and summarizing the relevant content.

2. There are many types of extracellular vesicles ranging from large to small. In addition to small exosomes, other types of extracellular vesicles should also be summarized for their functions and mechanisms.

3. Some sections of the article solely list the content of  other articles without providing a cohesive summary, which could be improved.

4. In the abstract section, the authors mentioned that the review is "based on ongoing clinical trials." However, as this review paper summarizes existing review papers rather than clinical trials, it is unclear why this statement is included.

5. In the introduction section, the authors stated, "To identify the most current significant publications in the PubMed database, we utilized the keywords 'extracellular vesicles, gastric cancer'. The only papers that met the inclusion requirements were from 2023–2024, as previously indicated." However, the references cited in this article are most review articles, and those from 2023-2024 primarily summarize data from earlier years. If the goal is to produce an updated review, should the focus be mainly on new research findings from the period of 2023-2024?

6. What is the connection between the title "The Importance of Extracellular Vesicle Screening in Gastric Cancer" and the section on "Nanovesicles role" within the article's content?

7. The title for section 4 does not effectively summarize the paragraph's content.

8. In section 5, the title is vague, and the content is not well-organized. Could you categorize and list subheadings for better clarity?

9. In the author contributions section, this article only provides a review of the articles. Why did some authors contribute to the methodology, formal analysis, and data curation?

Comments on the Quality of English Language

In this review paper, the authors reviewed EVs of gastric cancers. The review may help to understand the function and mechism of gastric cancer EVs but there are some suggestions for improving the work, it cannot be accepted at this state.

 1. As a review paper, it would be beneficial to include summaries with figures and tables to assist readers in better grasping the key points and summarizing the relevant content.

2. There are many types of extracellular vesicles ranging from large to small. In addition to small exosomes, other types of extracellular vesicles should also be summarized for their functions and mechanisms.

3. Some sections of the article solely list the content of  other articles without providing a cohesive summary, which could be improved.

4. In the abstract section, the authors mentioned that the review is "based on ongoing clinical trials." However, as this review paper summarizes existing review papers rather than clinical trials, it is unclear why this statement is included.

5. In the introduction section, the authors stated, "To identify the most current significant publications in the PubMed database, we utilized the keywords 'extracellular vesicles, gastric cancer'. The only papers that met the inclusion requirements were from 2023–2024, as previously indicated." However, the references cited in this article are most review articles, and those from 2023-2024 primarily summarize data from earlier years. If the goal is to produce an updated review, should the focus be mainly on new research findings from the period of 2023-2024?

6. What is the connection between the title "The Importance of Extracellular Vesicle Screening in Gastric Cancer" and the section on "Nanovesicles role" within the article's content?

7. The title for section 4 does not effectively summarize the paragraph's content.

8. In section 5, the title is vague, and the content is not well-organized. Could you categorize and list subheadings for better clarity?

9. In the author contributions section, this article only provides a review of the articles. Why did some authors contribute to the methodology, formal analysis, and data curation?

Author Response

Comments and Suggestions for Authors

In this review paper, the authors reviewed EVs of gastric cancers. The review may help to understand the function and mechism of gastric cancer EVs but there are some suggestions for improving the work, it cannot be accepted at this state.

1.As a review paper, it would be beneficial to include summaries with figures and tables to assist readers in better grasping the key points and summarizing the relevant content.

We introduced Figure 1 and Tables 1, 2 and 3

  1. There are many types of extracellular vesicles ranging from large to small. In addition to small exosomes, other types of extracellular vesicles should also be summarized for their functions and mechanisms.

We presented a classification based on what you said: size, function, and mechanism.

  1. Some sections of the article solely list the content of  other articles without providing a cohesive summary, which could be improved.

We have improved the context of some studies. We have also improved the context of what we have presented by introducing three tables, in which we presented additional data!

  1. In the abstract section, the authors mentioned that the review is "based on ongoing clinical trials." However, as this review paper summarizes existing review papers rather than clinical trials, it is unclear why this statement is included.

We changed it to “In this study, we provide an update on EVs and gastric cancer based on ongoing review papers and clinical trials.”

  1. In the introduction section, the authors stated, "To identify the most current significant publications in the PubMed database, we utilized the keywords 'extracellular vesicles, gastric cancer'. The only papers that met the inclusion requirements were from 2023–2024, as previously indicated." However, the references cited in this article are most review articles, and those from 2023-2024 primarily summarize data from earlier years. If the goal is to produce an updated review, should the focus be mainly on new research findings from the period of 2023-2024?

We changed to: “We focused mainly on new research findings from the period 2023-2024”

  1. What is the connection between the title "The Importance of Extracellular Vesicle Screening in Gastric Cancer" and the section on "Nanovesicles role" within the article's content?

We deleted this section: "Nanovesicles role"

  1. The title for section 4 does not effectively summarize the paragraph's content.

We changed the title to “2. Bacterial extracellular vesicles”

  1. In section 5, the title is vague, and the content is not well-organized. Could you categorize and list subheadings for better clarity?

We introduced subheadings.

  1. In the author contributions section, this article only provides a review of the articles. Why did some authors contribute to the methodology, formal analysis, and data curation?

We deleted: the methodology, formal analysis, and data curation.

Reviewer 3 Report

Comments and Suggestions for Authors

1. Author should correct typo errors and grammatical mistake.

2. Reform abstract.

3. Use figure and tables.

4. cite the reference

  • https://doi.org/10.1186/s12964-023-01370-3

  • 5. rewrite the conclusion
    •  

    •  
  •  
  •  

Comments on the Quality of English Language

1. Author should correct typo errors and grammatical mistake.

Author Response

Comments and Suggestions for Authors

1.Author should correct typo errors and grammatical mistake.

We corrected

2.Reform abstract.

We rewrote the abstract:

Simple Summary: The incidence of gastric cancer (GC) is increasing. Many people receive an advanced diagnosis of GC. Furthermore, a large number of patients are surgically inoperable and are unresponsive to preoperative chemotherapy. Patients with GC do not currently have an immunotherapy option. Our study's objective was to provide insight into extracellular vesicles (EVs) and GC. EVs in GC can be studied, as this article explains. EVs screening, which is available for all patients, is an option to combat GC.

Abstract: Extracellular vesicles, or EVs, are membrane-bound nanocompartments produced by tumor cells. EVs carry proteins and nucleic acids from host cells to target cells, where they can transfer lipids, proteomes, and genetic material to change the function of target cells. EVs serve as reservoirs for mobile cellular signals. The collection of EVs using less invasive processes has piqued the interest of many researchers. Exosomes carry substances that can suppress the immune system. If the results of exosome screening are negative, immunotherapy will be beneficial for GC patients. In this study, we provide an update on EVs and GC based on ongoing review papers and clinical trials.

3. Use figure and tables.

- We introduced Figure 1 and Tables 1,2 and 3

4. cite the reference

  • https://doi.org/10.1186/s12964-023-01370-3
  • We introduced this references

5.rewrite the conclusion

We rewrote the conclusions

  • Conclusions
  • The two most studied types of EVs in the CG are bEVs and exosomes. Due to the heterogeneity of EVs, research on EVs in GC is possible. The potential applications of exosome screening in GC patients include early diagnosis and follow-up. As these GC patients are often diagnosed in advanced stages, the detection and screening of EVs could improve the diagnosis of GC and represent a step forward in gastroenterology medical clinics. We recommend screening for EVs in GC in the clinic, and funding should be available for screening.

Round 2

Reviewer 2 Report

Comments and Suggestions for Authors

The quality of the manuscript has improved significantly following the revisions. One minor suggestion is to utilize color for Figure 1 and enhance the font size within the figure.

Comments on the Quality of English Language

The quality of the manuscript has improved significantly following the revisions. One minor suggestion is to utilize color for Figure 1 and enhance the font size within the figure.

Author Response

Thank you for your appreciation. It has been hard work!

We redid the figure, increased the font, and colored the figure!